# Gut-Flora-Dependent Metabolite Trimethylamine-N-Oxide Promotes Atherosclerosis-Associated Inflammation Responses by Indirect ROS Stimulation and Signaling Involving AMPK and SIRT1

**DOI:** 10.3390/nu14163338

**Published:** 2022-08-15

**Authors:** Sa Zhou, Jiamin Xue, Jingbo Shan, Yingxiang Hong, Wenkang Zhu, Zhiyan Nie, Yujie Zhang, Nanxi Ji, Xuegang Luo, Tongcun Zhang, Wenjian Ma

**Affiliations:** 1Key Laboratory of Industrial Fermentation Microbiology of the Ministry of Education, College of Biotechnology, Tianjin University of Science and Technology, Tianjin 300457, China; 2Institute of Biology and Medicine, Wuhan University of Science and Technology, Wuhan 430081, China; 3Qilu Institute of Technology, Jinan 250200, China

**Keywords:** TMAO, ROS, inflammation, gut microbiota, choline, atherosclerosis

## Abstract

Trimethylamine-N-oxide (TMAO), a gut-microbiota-dependent metabolite after ingesting dietary choline, has been identified as a novel risk factor for atherosclerosis through inducing vascular inflammation. However, the underlying molecular mechanism is poorly understood. Using an in vitro vascular cellular model, we found that the TMAO-induced inflammation responses were correlated with an elevation of ROS levels and downregulation of SIRT1 expression in VSMCs and HUVECs. The overexpression of SIRT1 could abrogate both the stimulation of ROS and inflammation. Further studies revealed that AMPK was also suppressed by TMAO and was a mediator upstream of SIRT1. Activation of AMPK by AICAR could reduce TMAO-induced ROS and inflammation. Moreover, the GSH precursor NAC could attenuate TMAO-induced inflammation. In vivo studies with mice models also showed that choline-induced production of TMAO and the associated glycolipid metabolic changes leading to atherosclerosis could be relieved by NAC and a probiotic LP8198. Collectively, the present study revealed an unrecognized mechanistic link between TMAO and atherosclerosis risk, and probiotics ameliorated TMAO-induced atherosclerosis through affecting the gut microbiota. Consistent with previous studies, our data confirmed that TMAO could stimulate inflammation by modulating cellular ROS levels. However, this was not due to direct cytotoxicity but through complex signaling pathways involving AMPK and SIRT1.

## 1. Introduction

Atherosclerosis is one of the most common cardiovascular diseases (CVDs) in humans and the leading cause of death worldwide [1]. The high level of cholesterol and saturated fat consumption in meat is a well-established risk factor of atherosclerosis [2]. TMAO is derived from dietary choline and carnitine in red meat, eggs and fish and is metabolized by gut microbiota to trimethylamine (TMA). TMA can be oxidized to TMAO by hepatic flavin monooxygenases FMO3 [1,3]. Excessive accumulation of TMAO has been identified as a novel risk factor for various CVDs, such as atherosclerosis [1,3,4], heart failure [5,6], stroke [7] and hypertension [8]. 

Previous studies suggest that TMAO regulates the bile acid synthetic pathway and cholesterol metabolism to promote the development of atherosclerosis [1,3,9]. Studies also showed that TMAO affected the mitogen-activated protein kinase (MAPK) and nuclear factor kappa-B (NF-κB) signaling to contribute to vascular inflammation [10]. However, the precise molecular mechanism by which circulating TMAO promotes atherosclerosis remains to be established.

It has been observed that atherosclerosis is caused by the formation of atheromatous plaque in artery walls, which leads to vessel stenosis and failure of blood flow. Accumulating evidence indicates that inflammation plays a key role and transduces many known risk factors to affect atherosclerosis [11,12]. Inflammatory signaling alters the behavior of the intrinsic cells (endothelium and smooth muscle) in the artery wall and recruits further inflammatory cells that interact to promote lesion formation and complications. The prolonged and excessive inflammatory processes of the vascular wall leading to atherosclerosis often begin with inflammatory changes of the endothelium [13,14]. However, there remain many unanswered questions about the cause and mechanism of inflammatory biology in human atherosclerosis.

Oxidative stress followed by chronic inflammation is suggested as another possible reason for the initiation of atherosclerosis in coronary arteries. Excess production of reactive oxygen species (ROS) relative to antioxidant defense could be a pivotal mediator of atherosclerosis through three mechanisms, including the direct damage of the cellular and nuclear membranes, the interaction with endogenous vasoactive mediators in endothelial cells and the production of oxidized lipoproteins—especially ox-LDL [15,16]. Many factors can lead to the production of ROS. However, experimental data are still limited concerning the generation of ROS by gut-microbiota-dependent metabolites and their potential destructive effects on endothelium function.

In addition to the stimulation of ROS from direct cytotoxic events, it is also known that ROS can be elevated as important signaling molecules. Sirtuin 1 (SIRT1), an NAD+-dependent protein deacetylase that plays a pivotal role in various physiopathological processes, including gene transcription and the cellular metabolism, has been shown to be an important mediator for the inflammation response and oxidative stress [17,18,19,20]. 

The inhibition of SIRT1 with siRNA or pharmacological agents induced a significant increase in ROS levels, indicating a potential relationship between SIRT1 and ROS [21]. However, ROS can also enhance SIRT1 activity by activating AMP-activated protein kinase (AMPK) to increase cellular NAD+ levels [22]. SIRT1 regulates Sirt1/NOX and Sirt1/SOD to perform potential antioxidative stress activity in vascular EC [23]. SIRT1 activity improves through modulation of the NOX 4/NADPH oxidative subunit, thus, reducing the ROS formation. 

In the current study, using human umbilical vein endothelial cells (HUVECs), aortic vascular smooth muscle cells (VSMCs) and a mouse model, we studied the molecular mechanisms on how TMAO could contribute to atherosclerosis development by exploring the possible underlying links between TMAO, ROS, inflammation factors and the signaling pathways involving SIRT1 and AMPK.

## 2. Materials and Methods

### 2.1. Cell Culture and Treatment

VSMCs and HUVECs were from Chinese Academy of Sciences, Shanghai, China. Monolayer cultures of VSMCss were maintained in the DMEM (Gibco, Grand Island, NY, USA) and monolayer cultures of HUVECs were maintained in the DF/12 (Gibco, Grand Island, NY, USA), containing penicillin (100 U/mL), 100 μg/mL of streptomycin, supplemented with 10% of fetal bovine serum (FBS) at 37 °C in a humidified 5% CO_2_ incubator.

All experiments were performed on cells following three to six passages at 80–90% confluence. VSMCs and HUVECs were treated with different concentrations of TMAO (50, 100, 200, 500, 1000 and 2000 μM) for 24 h for cell viability assay. Where indicated, cells were treated with TMAO (1000 μM) supplemented with NAC (5 mmol/L) or AICAR (5 mM) for 24 h for measurement of ROS, western blotting and quantitative real-time PCR analysis.

### 2.2. Cell Viability Assay 

Cell viability was determined by MTT (3-(4,5-dimethylthiazol-2-yl)-2,5-diphenyltetrazolium bromide) assay. VSMCs and HUVECs were plated on 96-well microplates at a density of 3 × 10^4^ cells/mL. Cells were treated with 50, 100, 200, 500, 1000 and 2000 μM TMAO for 24 h. Thereafter, cells were washed with PBS followed by incubation with the MTT solution (5 mg/mL, Sigma, St Louis, MO, USA) for 4 h. The optical density was measured at 490 nm with the use of a Synergy™ four-plate reader (Bioteck, Castelnau-le-Lez, France). The absorbance is considered proportional to the number of surviving cells.

### 2.3. Measurement of Reactive Oxygen Species

VSMCs and HUVECs were seeded in six-well plates and treated under different conditions. After treatment, the culture plates were washed with chilled PBS. Then, 10 μM 2′,7′-dichlorodihydrofluorescein diacetate (DCFH-DA) (Beyotime, Chengdu, China) was added per well at 37 °C in 5% CO_2_ for 30 min. To quantify the level of ROS, cells were harvested, and the DCFH-DA fluorescence was observed with laser confocal microscope (Olympus, Tokyo, Japan) and measured by BD Accuri C6 flow cytometry (Becton Dickinson, East Rutherford, NJ, USA). All experiments were repeated at least three times.

### 2.4. Western Blotting Analysis

Control and drug-treated cells were lysed using RIPA buffer and total proteins were separated by SDS-PAGE followed by western blot. The primary antibodies used were IL-1β (1:1000, 16806-1-AP rabbit, Proteintech, Rosemont, IL, USA), IL-6 (1:1000, 21865-1-AP rabbit, Proteintech, IL, USA), TNF-α (1:1000, 60291-1-Ig mouse, Proteintech, IL, USA), NLRP3 (1:1000, YT5382 rabbit, ImmunoWay Biotechnology, Plano, TX, USA), MMP9 (1:1000, ab38898 rabbit, Abcam, Cambridge, MA, USA), NF-κb (1:1000, 8242T mouse, Cell Signaling Technology, Beverly, MA, USA), SIRT1 (1:500, YT4302 rabbit, ImmunoWay Biotechnology, Plano, TX, USA), AMPK (1:1000, 2535T mouse, Cell Signaling Technology, Beverly, MA, USA), β-Actin (1:5000, 38,074 mouse, Signalway Antibody, Greenbelt, MD, USA). 

The secondary antibodies used were IRDyeTM-800-conjugated anti-mouse (1:5000, 926-32210, Li-Cor Biosciences, Lincoln, NE, USA) or anti-rabbit IgG (1:5000, 926-68071, Li-OR Biosciences, Lincoln, NE, USA). Immunoreactivity was detected using an Odyssey Infrared Imaging System (Gene Company Limited, Chai Wan, Hong Kong). The density quantifications were analyzed by ImageJ software (NIH, Bethesda, MD, USA). All experiments were repeated at least three times.

### 2.5. Quantitative Real-Time PCR 

The total RNA was isolated from cells using Trizol reagent (Invitrogen, Carlsbad, CA, USA). Reverse-transcribed cDNA was synthesized with random primers. The primers used for quantitative real-time PCR to detect the relative expression of genes are listed in Table 1. 18S was used as the internal control. cDNA was used as templates for quantitative real-time PCR, which was conducted using SYBR Green Master Mix (Invitrogen) in a Bio-systems StepOneTM Real-Time PCR machine (Applied Biosystem, Foster City, CA, USA). Three independent experiments were performed, each with three replicates.

### 2.6. Animals and Group Allocation

Five-week-old male C57BL/6 mice (weight, 23–25 g; ChangSheng Biology, Benxi, China) were housed individually at 24 ± 2 °C under a 12/12 h light/dark cycle with free access to food and water for at least 7 days before the start of the experiments. Then, the mice were randomly allocated into five groups (*n* = 5 per group). One group of mice continued with a standard diet, while the other four groups were switched to a choline diet. 

The four groups were respectively fed with chow diet supplemented with 1% choline (choline group), chow diet supplemented with 1% choline and 1 mmoL/L TMAO (choline+TMAO group), chow diet supplemented with 1% choline and 150 mg/Kg NAC (choline+NAC group), chow diet supplemented with 1% choline and 1 × 10^8^ CFU/mL *Lactobacillus plantarum* CGMCC 8198 (LP8198) (choline+LP8198 group) [24,25]. 

The experiment was performed for 12 weeks. At the end of the treatment period, mice were fasted for 12 h before collection of blood and tissues for analysis. Animals were euthanized and killed by cervical dislocation followed by decapitation. Blood, liver and aorta samples were collected immediately, snap-frozen in liquid nitrogen and stored at −80 °C until use. All animal experiments were performed in accordance with the Guide for the Care and approved by the Ethics Review Board for Animal Studies of the Tianjin University of Science & Technology (approval number SWKL2021113002—11/30/2021).

### 2.7. Plaque Lesion Analysis

The collected aortic roots and liver section were fixed in 4% paraformaldehyde. The samples were embedded with OCT and cut into 4 μm sections. Hematoxylin and eosin (H&E) and oil-red-O were performed for atherosclerotic lesion analysis.

### 2.8. Lipid Profile Analysis

After mice were sacrificed, blood serum was obtained by centrifuge. Concentrations of TC, HDL-TC and LDL-TC serum were measured using commercial kits according to the instructions (Jiancheng, Nanjing, China), and the concentrations of TMAO in mice were measured in serum using commercial ELISA kits (Taijin, Tianjin, China).

### 2.9. 16 S rRNA Gene Sequencing

The samples were processed by Shanghai Majorbio Bio-Pharm Technology Co, Ltd. (Shanghai, China). The total DNA was extracted, amplified and sequenced according to standard procedures. Briefly, microbial DNA was extracted using the E.Z.N.A.^®^ Soil DNA Kit (Omega Bio-tek, Norcross, GA, USA) according to the manufacturer’s protocol. 

The DNA concentration was assessed using a Nanodrop (Thermo Scientific, Wilmington, DE, USA), and the quality was determined by agarose gel electrophoresis. Bacterial 16S rRNA gene sequence spanning the variable regions V3–V4 were amplified using the primer statistical analysis 338F(5′-ACTCCTAGGGAGGGCAGCAG-3′)-806R(5′-GGACATCHVGGGTWTTCTAAT-3′). The amplicons were then extracted from 2% agarose gels and further purified using the AxyPrep DNA Gel Extraction Kit (Axygen Biosciences, Union City, CA, USA) and quantified by Quanti-FluorTM-ST (Promega, Madison, WI, USA). Purified amplicons were pooled in equimolar amounts and subjected to paired-end sequencing (2 × 300) on an lllumina MiSeq platform.

### 2.10. Statistical Analysis

The experimental data were expressed using the mean ± standard deviation (SD), and the statistical comparison was made by one-way ANOVA (which was used for multiple comparisons), two-way ANOVA (which was used to assess the interaction between two treatments) and Student’s *t*-test (which determines the degree of statistical significance between two experimental groups). All statistical analyses were performed using SPSS 26.0 software. P values less than 0.05 are considered statistically significant and noted in the figure legends, and significant differences are indicated with indicated with *, # or + (*,#,+ = *p* < 0.05, **, ##, ++ = *p* < 0.01).

## 3. Results

### 3.1. TMAO-Induced Inflammation by Stimulating ROS in VSMCs and HUVEC

TMAO, a gut-microbiota-dependent metabolite after ingesting dietary choline and carnitine in red meat, eggs and fish, was identified as a novel and risk factor for atherosclerosis. However, the precise mechanism regarding how TMAO induces atherosclerosis remains elusive. To study the underlying mechanism, we first investigated whether TMAO may directly cause cytotoxicity. Upon different doses of TMAO (50, 100, 200, 500, 1000 and 2000 μM) treatment in vitro on VSMCs and HUVECs (shown in Figure 1A,B), the cell viability was not affected, indicating that TMAO does not exert cytotoxic effects even at relatively high concentrations that cannot be produced in vivo (Figure 1A,B). 

Among the many factors that can accelerate atherosclerosis process, the generation of ROS [26] and the release of inflammatory chemokines have been reported to play pivotal roles [11,12,27]. Therefore, we wondered whether TMAO and antioxidant N-acetyl-L-cysteine (NAC) may impact atherosclerosis by affecting ROS generation and/or some level of inflammatory response. When VSMCs and HUVECs were treated with 1000 μM TMAO and 1000 μM TMAO + 5 mmol/L NAC for 24 h, the cellular level of ROS was dramatically increased after TMAO treatment as detected by the fluorescent dye DCFH-DA with confocal microscopy and flow cytometry (Figure 1C–E).

An increased ROS level is known to be one of the most common triggers for inflammation responses. ROS scavengers could suppress inflammasome activation [28,29]. On the other hand, inflammasome activators can also enhance ROS production [30]. The robust mtROS production was reported to be a major source driving inflammasome activation [31,32]. To investigate whether inflammatory chemokines could be activated by TMAO in VSMCs and HUVECs, we determined the expression of IL-1β, IL-6 and TNF-α, which are commonly used indicators of inflammatory activation. 

Other inflammatory factors NF-κB, matrix metallo peptidase 9 (MMP-9) and the NLRP3 inflammasome (an IL-1β family cytokine-activating protein) were also examined after TMAO treatment for 24 h in VSMCs and HUVECs by western blot and quantitative real-time PCR. The results showed that TMAO treatment stimulated both the protein levels and mRNA levels of IL-1β, IL-6, TNF-α, NF-κB, MMP9 and NLRP3 (Figure 1F–M and Appendix A).

To ensure further whether inflammation responses were induced by ROS, the antioxidant NAC was employed. As shown in Figure 1C–E, when VSMCs and HUVECs were treated with TMAO supplemented with NAC (5 mmol/L) for 24 h, the up-regulation of ROS level was markedly decreased compared to treating with TMAO only. In addition, the expression of IL-1β, IL-6, TNF-α, NF-κB, MMP9 and NLRP3 were all decreased though to different extents (Figure 1F–M and Appendix A). These results suggest that TMAO induces inflammation via ROS in VSMCs and HUVECs.

### 3.2. TMAO-Induced ROS Generation and Upregulation of Inflammatory Cytokines Were Mediated by SIRT1 in VSMCs and HUVECs

ROS are known as signaling molecules to trigger important physiological functions. One of the signaling pathways participating in the regulation of ROS is SIRT1-associated cellular networks. SIRT1 plays a pivotal role in various physiopathological processes, including cellular metabolism, inflammation and oxidative stress [33,34]. To further understand the mechanism on how TMAO stimulated the ROS level and inflammation, we determined whether TMAO could affect the expression of SIRT1 in VSMCs and HUVECs. 

The VSMCs and HUVECs were treated with 1000 μM TMAO for 24 h, and then the protein and transcript level of SIRT1 were determined by western blot and quantitative real-time PCR. As shown in Figure 2A,B, both the mRNA and protein level of SIRT1 were evidently decreased following TAMO treatment in VSMCs and HUVECs (Figure 2A,B and Appendix A). 

To further determine whether altered gene expression of SIRT1 was responsible for the upregulation of ROS, VSMCs and HUVECs were transfected with the pCMV-Tag2B-SIRT1 (SIRT1) vector to overexpress SIRT1 (Figure 2C,D and Appendix A) and then exposed to TMAO. As shown in Figure 2E,F, overexpression of SIRT1 in TMAO-treated VSMCs and HUVECs reduced the ROS levels comparing to that of without SIRT1 overexpression. 

Furthermore, the expression of major inflammatory cytokines TNFα, IL-1β and MMP9 were all inhibited in VSMCs and HUVECs when SIRT1 was overexpressed (Figure 2G,H and Appendix A). These results indicate that SIRT1 is an upstream mediator causing TMAO-induced ROS production and inflammatory response. 

### 3.3. TMAO-Modulated SIRT1 Expression That Is Dependent on AMKP Activity in VSMCs and HUVECs

Previous studies showed that increasing ROS and cellular NAD+ levels can activate AMPK to enhance SIRT1 activity [22]. Therefore, we questioned whether AMPK may also play a role to impact the signaling interactions between TMAO and SIRT1. First, we investigated whether AMPK could be affected by the presence of TMAO in VSMCs and HUVECs. As shown in Figure 3A,B, TMAO treatment led to a significant decrease in the expression level of AMPK, revealing a suppressive effect on both the protein and mRNA level (Figure 3A,B and Appendix A). To further confirm the relationship between AMPK and TMAO inhibition of SIRT1, the AMPK activator 5-aminoimidazole-4-carboxamide-1-beta-D-riboside (AICAR) was employed. 

The VSMCs and HUVECs were treated with 1000 μM TMAO, 5 mM AICAR and/or combinations of both for 24 h. Then, the protein and mRNA level of SIRT1 were determined. Comparing to the significantly decreased expression level of SIRT1 when treated with TMAO alone, both the protein and mRNA level of SIRT1 were increased in VSMCs and HUVECs when treated with addition of AICAR (Figure 3C,D and Appendix A). These data indicate that activation of AMPK could abrogate the inhibition of SIRT1 by TMAO.

### 3.4. TMAO-Induced ROS Accumulation and Inflammatory Cytokine Release Via AMPK-SIRT1 Pathway in VSMCs and HUVECs

To further observe whether the inhibition of AMPK activity by TMAO was responsible for altered ROS levels and gene expression of inflammatory factors, VSMCs and HUVECs were treated with TMAO and AICAR to detect the levels of ROS and inflammatory factors. We first determined whether the stimulated ROS level and inflammatory factors by TMAO could be abrogated by additional activation of AMPK. Cells were treated with TMAO, AICAR and TMAO plus AICAR for 24 h, respectively. As shown in Figure 4A,B, the addition of AMPK activator AICAR alleviated the increase levels of ROS induced by TMAO in VSMCs and HUVECs. 

Moreover, the expression level of TNF-α, IL-1β and MMP9 induced by TMAO were all decreased in the presence of AICAR (Figure 4C,D and Appendix A). The results suggest that AMPK activation can alter TMAO-induced the increase level of ROS and inflammation. In combination with the data in Figure 2 and Figure 3, these data clearly indicate that AMPK is upstream of SIRT1 in relaying the impact of TMAO toward modulating the ROS level and inflammation response.

### 3.5. TMAO and Choline Feeding Promotes Atherosclerosis Development in C57BL/6 Mice

To further determine the role of TMAO in causing atherosclerosis in vivo, C57BL/6J mice were fed with chow diet, choline dieta and choline diet +TMAO for 12 weeks. We also explored whether ROS inhibitor NAC and probiotic screened in our laboratory could alleviate TMAO-induced atherosclerosis. Choline diet+NAC and Choline diet + 1 × 10^8^ CFU/mL LP8198 were also used to feed C57BL/6J mice for 12 weeks. We first determined the vivo level of TMAO in mice after feeding choline and TMAO containing food. As shown in Figure 5A, choline and choline supplemented with TMAO led to a marked increase in the levels of plasma TMAO in C57BL/6 mice. Interestingly, the increase of the TMAO level associated with choline feeding was significantly decreased by NAC and the probiotic LP8198 (Figure 5A).

To further explore the effects of choline, TMAO, NAC and LP8198 on glycolipid metabolic disorders, we measured the three blood lipid-related parameters: the total cholesterol (TC), high-density lipoprotein cholesterol (HDL-C) and low-density lipoprotein cholesterol (LDL-C) in plasma. Compared with the chow diet group, TC in the choline and choline supplemented with TMAO groups was significantly higher. Supplementing NAC and LP8198 in the diet, however, significantly prevented the elevation of TC (Figure 5B). 

Moreover, the changes of HDL-C and LDL-C levels were similar to those of TC in each group (Figure 5C,D). As TC, HDL-C and LDL-C are generally considered as causal factors of atherosclerosis [35,36,37], this study provides further evidence that TMAO is a risk factor for atherosclerosis. In addition, the suppression of the three blood- lipid parameters by antioxidant NAC and the probiotic LP8198 is likely due to their impacts on TMAO.

To quantify the plaque burden and lipid content, the liver and aortic root sections were used for H&E staining. Comparing to the clear hepatic cellular structure observed in the chow diet group, the liver tissue in the choline and choline plus TMAO groups showed the characteristic structures with accumulation of lipid droplets, infiltration of inflammatory cells and partly fuzzy hepatic lobules. On the other hand, the lipid droplets and lesion area in choline plus NAC and choline plus LP8198 groups were significantly weaker than that in choline and choline+TMAO groups (Figure 5E and Appendix A). 

In addition, Oil-Red-O staining revealed that a clear and complete cross section of aorta could be observed in the chow diet group, whereas an elevated lipid content around the aorta and a larger lesion area in the aorta was observed in the choline and choline+TMAO groups. Consistent with the other parameters, no lipid content around the aorta and a clear and complete of aorta was observed in the choline+NAC and choline+LP 8198 groups (Figure 5E and Appendix A).

These results indicate that choline and TMAO feeding could promote atherosclerosis development as evaluated by multiple indicators, such as plasma, liver and arteries, while NAC and LP 8198 can mitigate this development trend.

### 3.6. Effects of TMAO, NAC and LP8198 on Community Diversity and Richness of the Gut Microbiota in Choline-Fed Mice

TMAO is a metabolite, derived from dietary choline by gut microbiota. In order to explore whether high level of TMAO in vivo adversely affects gut microbiota, we evaluated the effects of TMAO, NAC and LP8198 on the gut microbiota, and the V3–V4 regions of 16S rRNA genes amplified from five groups of colonic content samples were sequenced. The community diversity and richness were calculated using the alpha diversity metric. As shown in Figure 6A–C, the Chao, Sobs and Ace index indicated that a choline diet caused a significant increase in the richness of the intestinal microbiome compared to the chow diet. 

However, the richness of the intestinal microbiome showed a slight, albeit not significant, decrease in the choline+TMAO group, choline+NAC group and choline+LP8198 group compare with in the choline diet group (Figure 6A–C). The Shannon diversity estimator was also higher in the choline diet groups than in the chow diet group. However, NAC and LP8198 in combination with choline did not affect the diversity of intestinal flora compared to choline alone (Figure 6D). These results suggest that a choline diet can increase the richness and diversity of intestinal flora in mice, while NAC and LP8198 had little effect on the intestinal flora. 

### 3.7. Principal Component Analysis (PCA) and Species Composition Analysis of Intestinal Flora in Choline, Choline+TMAO, Choline+NAC and Choline+LP 8198 Mice

In order to determine the overall structural differences (including the composition and abundance) of intestinal flora among different groups, PCA was performed based on the Bray–Curtis distance. The species composition was analyzed with a Venn diagram. The Venn diagram can more intuitively show the number of common or unique species in each intervention group, thus, explaining the similarity and difference of intestinal flora composition in each group to a certain extent. 

The results showed that the choline group and chow group were separated clearly and that the choline+TMAO, choline+NAC and choline+LP8198 groups deviated more or less from the choline group (Figure 7A). This indicated that the composition of the microbial community in the gut was changed when the mice were fed with choline and that TMAO, NAC and LP8198 had an effect on the gut microbiota of choline-fed mice. 

PCA analysis was also conducted on the choline group and choline+TMAO/choline+NAC/choline+LP8198 groups (Figure 7B–D). The results showed that there is some overlap between choline diet alone group and choline+TMAO/choline+NAC group (Figure 7A–C), while there was dispersed distribution between the choline+LP8198 group and choline diet group (Figure 7A,D). This suggests that the effects of LP8198 on the gut microbiota of choline-fed mice were more pronounced.

The species diversity values of the choline group, choline+TMAO group choline+NAC group and choline +8198 were higher than with the chow diet (Figure 7E). The results indicated that a choline diet can increase the species diversity of intestinal flora in mice. Moreover, the species composition results of the choline group and choline +NAC group were similar; however, the species composition results of the choline group, choline+TMAO group and choline+LP8198 group were different (Figure 7E–H). The results suggested that the bacterial community composition of the choline diet group was obviously different from the chow diet group and that TMAO, NAC and LP8198 could also alter the bacterial community composition.

### 3.8. Effects of Choline Diet, NAC and LP8198 Exposure on Composition of the Intestinal Microbiome

In order to evaluate the influence of each group on the specific composition of intestinal flora, a stacked histogram was used to analyze the intestinal flora of each group at the portal level. The two phylum Bacteroidetes and Firmicutes with the highest abundance in mice colon content flora are called dominant phylum and counted.

At the phylum level, Bacteroidetes and Firmicutes were the predominant phylum in the intestinal microbiome of mice, with a total abundance of nearly 90%, followed by Verrucomicrobia, Campilobacterota and Actinobacteriota. The ratio of Firmicutes/Bacteroides is called F/B ratio, which can be used to evaluate the severity of intestinal flora imbalance [38]. These microbes and the F/B ratio showed differences after exposure to choline, choline+TMAO, choline+NAC and choline+LP8198. 

Among these, the relative abundance of Firmicutes increased significantly with the choline diet, especially in the choline+TMAO group (Figure 8A). Moreover, the F/B ratio of choline diet groups was increased compared with chow group, especially in the choline+TMAO group, while the F/B ratio of choline+NAC and choline+LP8198 groups had no significant effect compared to that of choline alone group. Therefore, the choline diet induced the ecological imbalance of intestinal flora in mice, and NAC and LP8198 could relieve the imbalance of intestinal flora (Figure 8B). 

## 4. Discussion

In the present study, we found: (1) TMAO induced the inflammation response in VSMCs and HUVECs, which may contribute to the occurrence of atherosclerosis; (2) a high level of circulating TMAO induced by excessive choline intake can cause atherosclerosis and affected the community composition and structure of the gut microbiota in the colon of mice; (3) through repressing the expression of AMPK and SIRT1 and stimulating oxidative stress, TMAO activated the NF-κB/TNFα/IL1 pathway in VSMCs and HUVECs; and (4) NAC and probiotics had little effect on the composition and structure of the gut microbiota community but alleviated atherosclerosis induced by TMAO. 

Taken together, the current study suggests that elevated circulating TMAO levels caused by intake-choline may deteriorate the liver and aortic root, which is associated with the repression of AMPK and SIRT1 expression, increased ROS level and activation of the inflammation response pathway. Recently, the association of health-gut microbiota and their metabolites has attracted great interest [39]. The gut microbiota has been shown to affect various diseases through their metabolites [9]. 

TMAO, a gut-microbiota-dependent metabolite, could induce various cardiovascular disease. The level of circulating TMAO is determined by several factors, including diet, gut microbial flora, liver FMO enzymes and kidney function. In this study, we found that excessive intake of choline from red meat and eggs induced the increase level of circulating TMAO and thus affected the gut microbiota composition, TC, HDL-C and LDL-C promoted atherosclerosis development in mice (Figure 5, Figure 6, Figure 7 and Figure 8). However, the exact mechanisms remain to be elucidated.

In the present study, we showed that TMAO triggered vascular inflammation via ROS and signaling involving AMPK and SIRT1 in vitro. We first found that TMAO treatment had no cytotoxicity to cells (Figure 1). These results are consistent with previous reports from other labs [9,40]. Researchers confirmed that physiological levels of TMAO can promote inflammation via mitogen-activated protein kinase (MAPK) and the NF-κB signaling pathway in vascular endothelial and smooth muscle cells [10]. 

Research reported that TMAO induces vascular inflammation by activating the NLRP3 inflammasome [40]. It was also reported that TMAO dose-dependently induced the expression of TF and vascular cell adhesion molecule (VCAM)1 in human microvascular endothelial cells (HMECs), which were mediated arterial thrombosis [41]. Our results supported that TMAO induces inflammation response by IL-1β, IL-6, TNF-α, NF-κB, MMP9 and NLRP3 and deteriorates the liver and aortic root. These results support a role for TMAO in the activation of inflammatory pathways in vascular of vivo and vitro, leading to AS.

Chen et al. [40] reported that TMAO treatment induced both total ROS and mtROS levels to a significant extent in a dose- and time-dependent manner. Sun et al. [42] showed that intracellular ROS and MDA in the different concentrations of TMAO stimulus groups were clearly higher than in a control group, while the levels of SOD were significantly decreased by TMAO treatment in HUVECs. 

Wu et al. [43] proved that TMAO promotes apoE−/− mice atherosclerosis by inducing vascular endothelial cell pyroptosis via the SDHB/ROS pathway. TMAO induces ROS generation through NADPH oxidase activation in primary trophoblasts and HTR-8/SVneo cells [44]. Our study demonstrated that TMAO dramatically increased cellular ROS level (Figure 1). These studies have demonstrated that TMAO induces ROS production in a variety of cells both in vitro and in vivo.

Previous studies showed that ROS production induced by TMAO could be significantly inhibited by apocynin (an NADPH oxidase inhibitor) but not by rotenone (an inhibitor of mitochondrial respiratory chain complexes) or allopurinol (a xanthine oxidase inhibitor), which demonstrated that NADPH oxidase activation was the main source of TMAO-induced ROS production [44]. However, Chen et al. found that mtROS scavenger-TEMPO markedly inhibited TMAO-induced activation of the NLRP3 inflammasome and decreased the expression of inflammatory chemokines IL-1β, ICAM-1 and MMP-9 [40]. 

NAC, as a pharmacological inhibitor of ROS, could reverse TMAO-induced abnormity levels of ROS and inflammatory cytokine IL-1β and IL-18 and regulate VCAM-1 gene transcription and expression in HUVECs and human endothelial cells [42,45,46]. Our results that NAC inhibited TMAO-induced ROS generation followed by the decreased expression of inflammatory cytokines were consistent with these studies. NAC is a pro-drug and a non-enzymatic antioxidant serving as an antidote for paracetamol poisoning. Our present finding that NAC could alleviate the effects of TMAO suggest that it may be potentially helpful for atherosclerosis as well.

In the present study, we found that SIRT1 is an upstream mediator causing TMAO-induced ROS production and inflammatory response. There has been abundant evidence that SIRT1 is an important regulator of inflammation response and plays a key role in regulating the innate immune response [47,48,49,50,51]. Activation of SIRT1 can deacetylate NF-κB and bind to the promoter of TNFα, inhibiting the expression of NF-κB, TNFα and other pro-inflammatory cytokines in endotoxin-tolerance responses [51,52]. 

Activation of SIRT1 reduces ROS generation through stimulating the activity of the NOX4/NADPH oxidase, which then regulates cellular oxidative stress [53,54]. In this study, we further demonstrated that TMAO induces ROS and the inflammation response by impairing the expression of SIRT1, which could be restored via the overexpression of SIRT1. Earlier studies showed that AMPK activation could inhibit oxidative stress [55] and cause anti-inflammatory effects by inhibiting TNF-α, IL-1β, monocyte chemotactic protein (MCP)-1 and intercellular adhesionmolecule (ICAM)-1 expression in retinal pigment epithelium (RPE) cells [56]. Consistent with these studies, the present study further demonstrated that the inhibition of TMAO on SIRT1 may also be regulated by the AMPK pathway (Figure 3 and Figure 4). 

AMPK and SIRT1 play important roles in a variety of physiological processes, including regulating energy expenditure [22], attenuating oxidative stress-induced apoptosis [57], regulating hyperglycemia-induced cell death [58], attenuating ethanol-induced liver injury [59], relieving non-alcoholic fatty liver disease [60] and activating lysosomal function in the brain [61]. Here, we demonstrate that TMAO could regulate the AMPK-SIRT1 pathway, which is responsible for TMAO-induced oxidative stress and inflammation effects (Figure 1, Figure 2, Figure 3 and Figure 4).

In vivo, this study is consistent with the previous reports that TMAO can be produced from choline through the metabolism and that the exogenous intake of TMAO is also capable of directly increasing the in vivo TMAO level [40,62,63]. The exogenous intake of choline and TMAO induced increased levels of TC, HDL-C and LDL-C, which are considered as causal factors in atherosclerosis [35,36,37]. Interestingly, the increased levels of TMAO and TC associated with choline feeding were significantly decreased by NAC and the probiotic LP8198, and the levels of HDL-C and LDL-C were also decreased by LP8198 (Figure 5). 

A growing number of studies have shown that the combination of microbiome and its associated metabolites could be helpful in understanding the possible mechanisms of disease development [64,65]. A previous study reported that berberine attenuates choline-induced atherosclerosis by inhibiting TMA and TMAO production via regulating the gut microbiome [66]. 

In our study, the gut-flora-dependent metabolite TMAO was further analyzed for its effect on the gut flora. The findings extended the relationship analysis between the gut flora and metabolites. By self-screening, the probiotic LP8198 was added to assess the effects of TMAO. To our knowledge, this is the first study where self-screening probiotics was used to mitigate the effects of TMAO. Moreover, LP8198 was found to alleviate TMAO-induced atherosclerosis.

## 5. Conclusions

In summary, this study demonstrated that TMAO is associated with the development of atherosclerosis in vitro and in vivo. In vitro, we demonstrated that TMAO regulated ROS stimulation and AMPK and SIRT1 signaling to induce inflammation responses in VSMCs and HUVECs, which may contribute to promoting atherosclerosis (Figure 9). Moreover, the antioxidant NAC can inhibit the production of ROS and alleviate the inflammatory responses in VSMCs and HUVECs. In vivo, choline and TMAO promote the development of atherosclerosis. NAC and LP8198 also decelerate TMAO-induced atherosclerosis and affect the gut microbiota of C57BL/6J mice.

## Figures and Tables

**Figure 1 nutrients-14-03338-f001:**
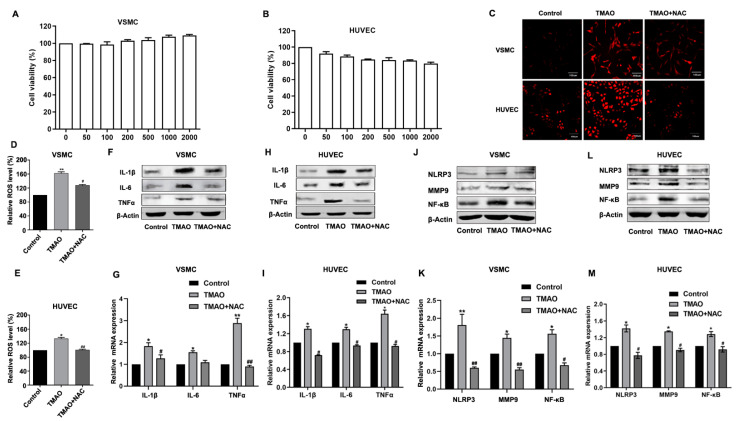
TMAO-induced inflammation via ROS in VSMCs and HUVECs. (**A**,**B**). VSMCs (**A**) and HUVECs (**B**) were incubated with different concentrations of TMAO (100, 200, 500, 1000 and 2000 μM) for 24 h. Thereafter, the cell viability was determined. (**C**–**E**) VSMCs and HUVECs were treated with 1000 μM TMAO for 24 h. The total ROS levels were determined by staining with DCFH-DA. Then, the ROS levels were observed by confocal microscopy (**C**) and measured by flow cytometry (**D**,**E**). (**F**–**M**) The protein levels (**F**,**H**,**J**,**L**) and relative mRNA levels (**G**,**I**,**K**,**M**) of inflammation-related markers IL-1, IL-6 and TNFα (**F**–**I**) and other inflammatory factors NLRP3, MMP9 and NF-κb (**J**–**M**) in VSMCs (**F**,**G,J,K**) and HUVECs (**H**,**I,L,M**) were detected by western blot and quantitative real-time PCR. The values are the mean ± SD; *n* = 3; one-way ANOVA followed by a Tukey post hoc test. * *p* < 0.05; ** *p* < 0.01 vs. Control; # *p* < 0.05; ## *p* < 0.01 vs. TMAO group.

**Figure 2 nutrients-14-03338-f002:**
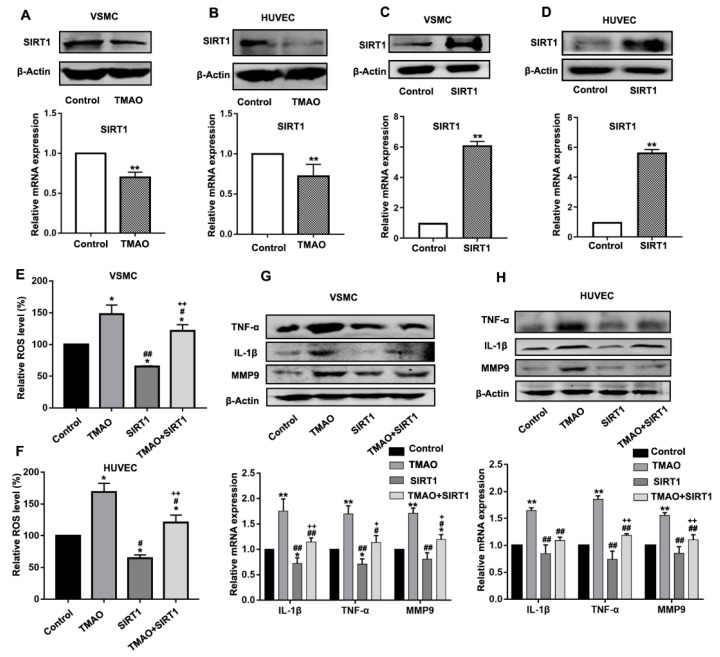
TMAO increased the ROS level and inflammatory factors by inhibiting SIRT1 expression. (**A**–**D**).The protein level (upper panels) and transcript level (lower panels) of SIRT1 were determined in VSMCs (**A**,**C**) and HUVECs (**B**,**D**) treated with 1000 μM TMAO (**A**,**B**) and transfected with pCMV-Tag 2B- SIRT1 plasmid (**C**,**D**) for 24 h by western blot and quantitative real-time PCR. (**E**,**F**) VSMCs (**E**) and HUVECs (**F**) were treated with 1000 μM TMAO, SIRT1 vector and 1000 μM TMAO plus SIRT1 vector. The total ROS levels were determined using fluorescence intensity of DCFH-DA staining by flow cytometry. (**G**,**H**) The protein level (upper panels) and transcript level (lower panels) of inflammation-related markers IL-1β, TNFα, MMP9 were determined in VSMCs (**H**) and HUVECs (**F**) treated with 1000 μM TMAO, SIRT1 vector and 1000 μM TMAO plus SIRT1 vector for 24 h. Values are the mean ± SD; *n* = 3; two-way ANOVA followed by a Tukey post hoc test. * *p* < 0.05; ** *p* < 0.01 vs. Control; # *p* < 0.05; ## *p* < 0.01 vs. TMAO group; + *p* < 0.05; ++ *p* < 0.01 vs. SIRT1 group.

**Figure 3 nutrients-14-03338-f003:**
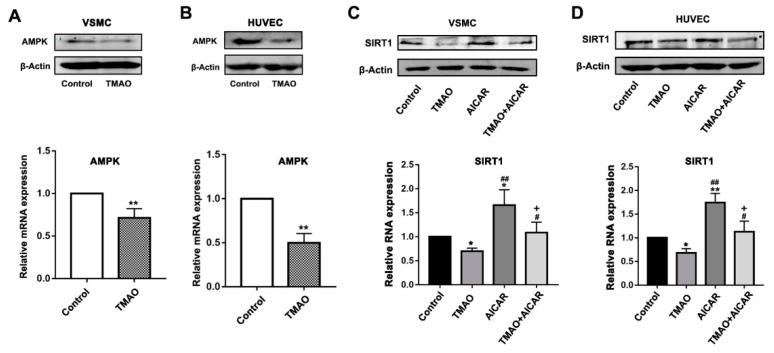
TMAO downregulates the SIRT1 level through AMPK in VSMCs and HUVECs. (**A**,**B**). The protein level (upper panels) and relative mRNA level (lower panels) of AMPK were detected after 1000 μM TMAO treatment for 24 h by western blot and quantitative real-time PCR in VSMCs (**A**) and HUVECs (**B**). Values are the mean ± SD; *n* = 3, Student’s *t*-test. ** *p* < 0.01 vs. Control. (**C**,**D**) The protein level (upper panels) and relative mRNA level (lower panels) of SIRT1 was detected after 1000 μM TMAO, 5 mM AICAR and 1000 μM TMAO plus 5 mM AICAR treatment for 24 h by western blot and quantitative real-time PCR in VSMCs (**C**) and HUVECs (**D**). Values are the mean ± SD; *n* = 3; two-way ANOVA followed by a Tukey post hoc test. * *p* < 0.05; ** *p* < 0.01 vs. Control; # *p* < 0.05, ## *p* < 0.01 vs. TMAO group; + *p* < 0.05 vs. the AICAR group.

**Figure 4 nutrients-14-03338-f004:**
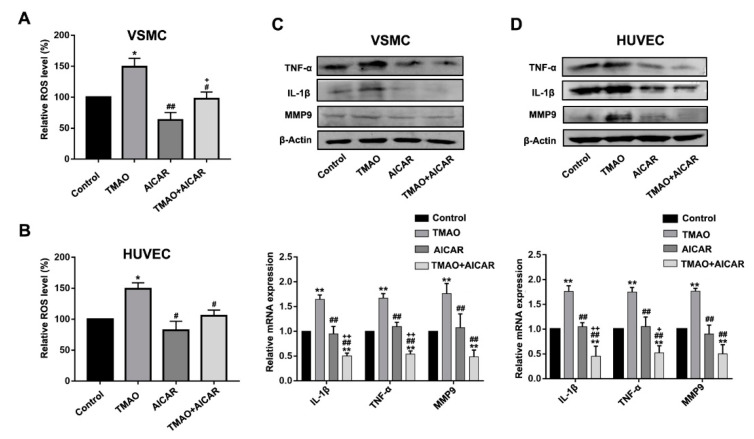
TMAO increased the ROS level and inflammatory factors by inhibiting AMPK levels in VSMCs and HUVECs. (**A**,**B**) The total ROS levels were determined after 1000 μM TMAO, 5 mM AICAR and 1000 μM TMAO plus 5 mM AICAR treatment for 24 h using fluorescence intensity of DCFH-DA staining by flow cytometry in VSMCs (**A**) and HUVECs (**B**).(**C**,**D**) The protein level (upper panels) and relative mRNA level (lower panels) of TNFα,IL-1β and MMP9 were detected by western blot and real-time PCR in VSMCs (**C**) and HUVECs (**D**). Values are the mean ± SD; *n* = 3; two-way ANOVA followed by a Tukey post hoc test. * *p* < 0.05; ** *p* < 0.01 vs. Control; # *p* < 0.05; ## *p* < 0.01 vs. TMAO group; + *p* < 0.05; ++ *p* < 0.01 vs. the AICAR group.

**Figure 5 nutrients-14-03338-f005:**
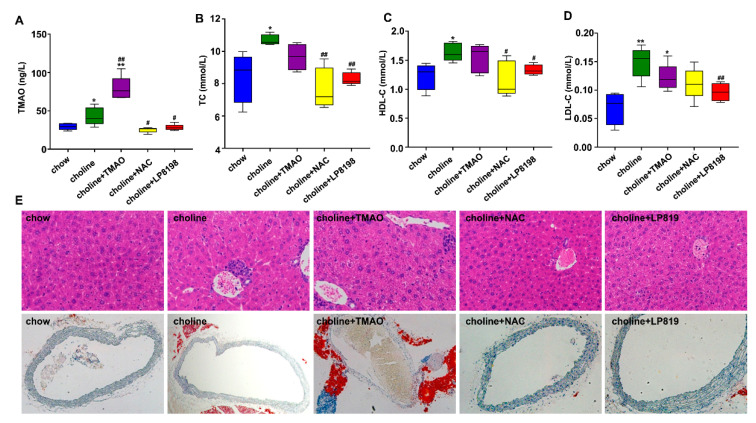
Choline diet and TMAO aggravates the occurrence of atherosclerosis in C57BL/6 mice. (**A**) TMAO levels were detected at indicated time points by ELISA kits. (**B**) Serum total cholesterol (TC). (**C**) Serum high-density lipoprotein cholesterol (HDL-C). (**D**) Serum low-density lipoprotein cholesterol (LDL-C). (**A**–**D**). Values are the mean ± SD; *n* = 5 per group, two-way ANOVA followed by a Tukey post hoc test. * *p* < 0.05; ** *p* < 0.01 vs. the chow group; # *p* < 0.05; ## *p* < 0.01 vs. the choline group. (**E**) Representative images of H&E staining at liver and oil-Red O staining at aortic root in C57BL/6 mice fed with choline diet with or without TMAO, NAC, LP8198 for 12 weeks.

**Figure 6 nutrients-14-03338-f006:**
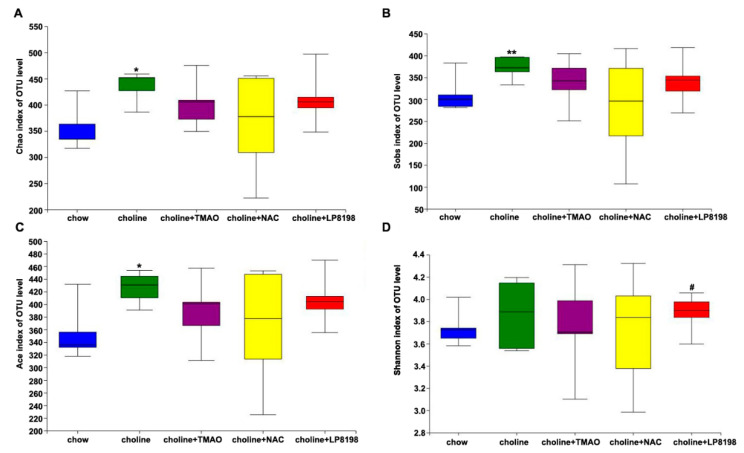
Alpha diversity of bacterial communities. Estimation of bacterial richness (**A**–**C**) and diversity (**D**) is visualized using boxplots for chow (blue), choline (green), choline+TMAO (purple), choline+NAC (yellow) and choline+LP8198 (red) groups. The boxplots show the median, quartile, smallest and largest observations. *n* = 5 per group, the Wilcoxon rank-sum test was performed to determine statistical significance of alpha diversity analyses. * *p* < 0.05; ** *p* < 0.01, vs. the chow group. # *p* < 0.05, vs. the choline group.

**Figure 7 nutrients-14-03338-f007:**
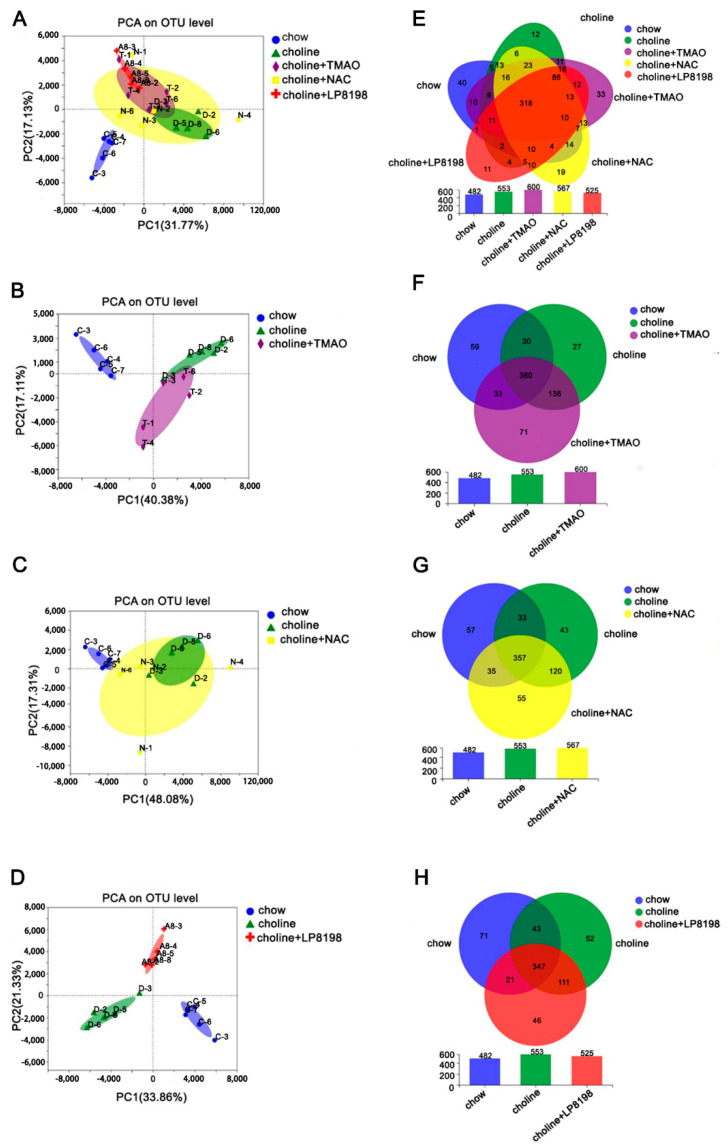
PCA and species composition analysis of the intestinal flora in C57BL/6 mice. (**A**) PCA analysis of all samples. (**B**) PCA analysis of chow-diet group, choline-diet group and choline-diet+TMAO group. (**C**) PCA analysis of the chow-diet group, choline-diet group and choline-diet+NAC group. (**D**) PCA analysis of chow-diet group, choline-diet group and choline-diet+LP 8198 group. (**E**) Venn diagram analysis of all samples. (**F**) Venn diagram analysis of chow-diet group, choline-diet group and choline-diet+TMAO group. (**G**) Venn diagram analysis of chow-diet group, choline-diet group and choline-diet+NAC group. (**H**) Venn diagram analysis of chow-diet group, choline-diet group and choline-diet+LP 8198 group. *n* = 5 per group.

**Figure 8 nutrients-14-03338-f008:**
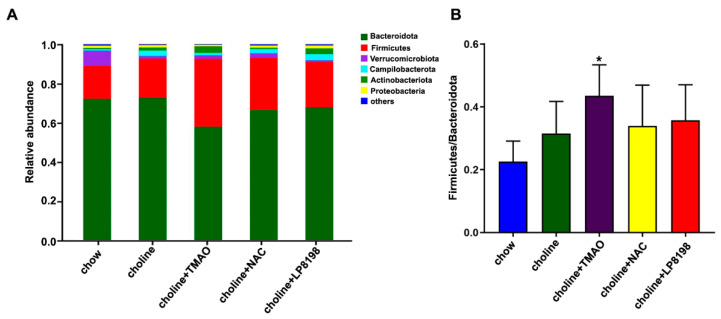
A choline diet alters the composition of the intestinal microbiome. (**A**) Composition and relative abundances of bacterial phyla in different groups. (**B**) The ratio of Firmicutes/Bacteroides in different groups. Values are the mean ± SD; *n* = 5 per group; two-way ANOVA followed by a Tukey post hoc test. * *p* < 0.05 vs. the chow group.

**Figure 9 nutrients-14-03338-f009:**
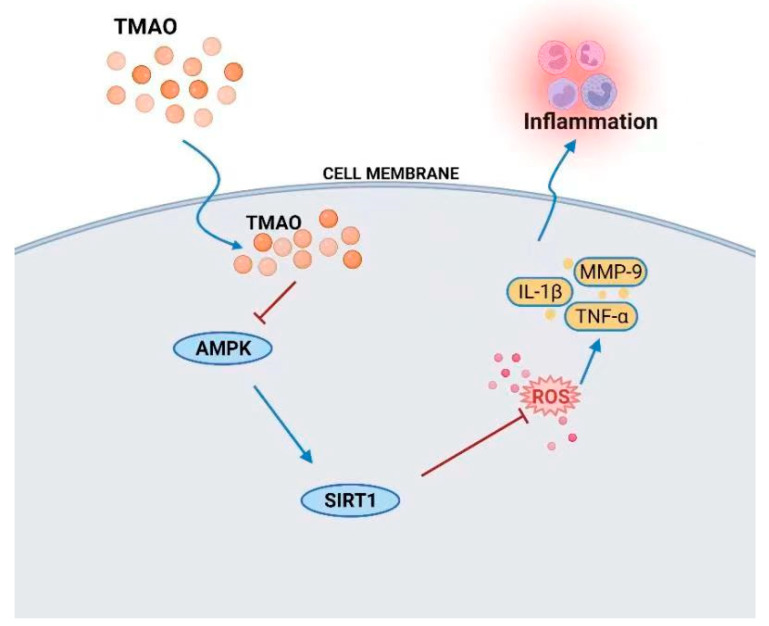
TMAO-mediated inflammation via ROS and the AMPK/SIRT1-signaling pathway.

**Table 1 nutrients-14-03338-t001:** Primer sequences for the target genes.

GenBank Accession Number	Gene	Sequences (5′-3′)
NR_146119.1	18S	Forward CAGCCACCCGAGATTGAGCA
		Reverse TAGTAGCGACGGGCGGTGTG
NM_000576.3	IL-1β	Forward GCTCTCCACCTCCAGGGACA
		Reverse AGGCCCAAGGCCACAGGTAT
NM_000600.5	IL-6	Forward CACACAGACAGCCACTCACC
		Reverse CCAGTGCCTCTTTGCTGCTT
NM_000594.4	TNF-α	Forward TCTGGCCCAGGCAGTCAGAT
		Reverse ATTGGCCAGGAGGGCATTGG
NM_001079821.3	NLRP3	Forward GACCCAAACCCACCCATCCA
		Reverse AGCGGTCCTATGTGCTCGTC
NM_004994.3	MMP9	Forward ACAGCCGGGACGCAGACATC
		Reverse GGCCAGGAGGAAAGGCGTGT
NM_001165412.2	NF-κB	Forward CAGCAGATGGCCCATACCTT
		Reverse TTTTCACTAGAGGCACCAGG
NM_001142498.2	SIRT1	Forward GCTGGAACAGGTTGCGGGAA
		Reverse GCTGGGCACCTAGGACATCG
NM_001355028.2	AMPK	Forward ACAGCCGAGAAGCAGAAACA
		Reverse CTTCACTTTGCCGAAGGTGC

## Data Availability

The data presented in this study are available within the article.

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
