# Peer review of "Gut-Flora-Dependent Metabolite Trimethylamine-N-Oxide Promotes Atherosclerosis-Associated Inflammation Responses by Indirect ROS Stimulation and Signaling Involving AMPK and SIRT1"

_nutrients, 2022, doi:10.3390/nu14163338_

Round 1
Reviewer 1 Report
The authors nicely showed that Choline/TMAO induces ROS-production and signaling via AMPK and SIRT1. The authors also state that these mechanisms are (at least in part) responsible for promoting atherosclerosis. In my opinion, the investigation of the pro-atherogenic effect is a weakness of the paper. The title suggests that ROS/AMPK/SIRT1 are responsible for the atherogenicity of TMAO. In contrast, 90% of the paper deal with cell culture. Therefore, in my opinion this statement is too ambiguous. The pictures of the aortic lesions in Fig. 5E are not convincing. Especially such lesions in C57BL/6 mice after 12wks. Please provide all pictures from all mice (also n=? is missing in the methods for each group) as a supplementary file and quantify the effect – otherwise I would recommend to focus in this paper on your good cell culture results instead of showing a vague effect on atherosclerosis.
It has been shown that in human microvascular endothelial cells TMAO dose-dependently induced expression of TF and vascular cell adhesion molecule (VCAM)1 – this mediated arterial thrombosis (please discuss DOI: 10.1093/cvr/cvab263). Does NAC-treatment also influences TF/VCAM-1 expression?
Reviewer 2 Report
Zhuo et al. study the effects of TMAO on the mechanism of ROS formation and inflammatory signaling in epithelial cells in vitro and the effects of choline-dependent TMAO formation and associated atherosclerosis in vivo. They find, that TMAO treatment reduces AMPK and SIRT1 expression in HUVECS and VSMCs and increases ROS formation in an AMPK and SIRT1 dependent manner. They further show, that presence of the anti-oxidant NAC reduces the ROS induced inflammatory response. Further in in vivo studies, the provide evidence, that choline feeding stimulates TMAO production and causes increases in plasma lipid as well as aortic root lipid levels, which can be alleviated by either NAC or LP8189 co-treamtment. They also investigate the effects of choline enriched diet plus probiotics or anti-oxidant supllementation on the intestinal gut flora, which the find altered after cholin-treatment but not so much affect by either NAC or LP8189 co-treatment. While the study focuses on a highly relevant topic and also includes a pro-biotic compound as intervention, many of the findings the authors present as novel have indeed been described before. In addation to alterations in the wording regarding the novelty of the study, there are several other points, that have to be addressed by the authors:
Major:
Methods: Please also add a section explaining the TMAO treatment as well as the TMAO+NAC treatment
Line180-182: Please also indicate the use of NAC already here before you introduced the expression of inflammatory cytokines.
Figure 1: why do you skip G in the figure designation, usually one would expect E, F, G, H instead of E,F, H,I. Also, please indicate in the figure legend, that the upper panel refers to the protein levels and the lower panel to the gene expression data. Is the any quantification of the western blots available? I highly recommend showing that (maybe in a suppl. Figure) especially as you only show one sample per condition.
Figure 2: please indicate in the figure legend, that the upper panel refers to the protein levels and the lower panel to the gene expression data. In the figure legend it reads STIR1 instead of SIRT 1 several times, please correct. In the figure legend, it says: “VSMCs (E) and HUVECs (F) was treated”, this should be changed into “VSMCs (E) and HUVECs (F) were treated.
Lines 226 and follows: Is there also a significant effect of SIRT1 overexpression only on ROS levels or gene/ protein expression? This should also be tested, using a two-way ANOVA since you compare two treatment regimen (SIRT1 overexpression and TMAO). Also, this is not mentioned at all the in text. A thorough description of all the data is required. In the same line as before: please show a quantification of the Western blot from >1 sample.
Line 242 an follows related to figure 3: Compared to figure 2, where you show, that TMAO treatment significantly decreases SIRT1 expression, this effect is not evident anymore? Or do you just do not show it?
Line 242: The heading of the paragraph is overstated. Maybe you should be less offensive and just state, that TMAO modulates SIRT1 expression dependent on AMKP activity, as AICAR is an AMPK activator. In the same line, before you argue, that TMAO affects SIRT1 expression through AMPK, you may first show an effect of TMAO on AMPK, as you do in figure 4.
Line 262-263: “To further confirm whether the inhibition of SIRT1 by TMAO was regulated by 262 AMPK. We also… ” should be “To further confirm whether the inhibition of SIRT1 by TMAO was regulated by 262 AMPK, we also…
Figure 4: as before: please indicate in the figure legend which panel refers to protein and gene expression and add a quantification of the protein levels (at least in a suppl. Figure with >n=1).
Line 270-271: Although you stated, that “AICAR alleviated the increase levels of ROS induced by TMAO in VSMCs and HUVECs” you do not show this in the figure as there is no statistical comparison. As before, I suggest to perform a two-way ANOVA.
Additionally, what I find a bit confusing is that, the AICAR TMAO effects on ROS levels (TMAO treatment increases ROS level (4C, 4D), AICAR alone seems to reduce ROS level compared to control (4C,4D), and TMAO induced ROS levels is attenuated by AICAR treatment) do not match the gene and protein expression data. Here it seems that AICAR alone treatment has no effects (compared to control) while combined AICAR and TMAO treatment dramatically reduce pro-inflammatory signaling, to a way greater extent than ROS levels would suggest.
Figure 5: Please indicate in the figure legend what the significance mean. Also here I highly recommend to perform two-way ANOVA.
Neither the method nor the results section and figure legend gives information about the mice used per group, please add.
Line 324: While the authors state in the figure legend of Figure 5 that “TMAO levels were 324 detected at indicated time points by HPLC-MS/MS.” in the methods section they mention that “concentrations of TMAO in mice were 146 measured in serum using commercial ELISA kits (Taijin, Tianjin, China) (line 146-147”. This is conflicting to me.
Figure 5 E: One picture each (n=1) showing the liver and aortic root is data is really weak. The exemplary picture do require proper quantification.
Why did the authors use WT mice in order to perform an atherosclerosis study? Better would have been the use of a common atherosclerotic mouse model such as ApoE ko and LDlr ko mice.
Line 331: The sentence “TMAO, as a metabolite, derived from dietary choline by gut microbiota.” Should be rewritten as “TMAO, is a metabolite, derived from dietary choline by gut microbiota.”
Line 331 – 334: The punctuation of sentences has to be improved.
Line 338: the authors state that “intestinal microbiome showed a slight decrease in choline +TMAO group” however according to figure 6 this is not significant. Accordingly, the wording should be changed into “showed a slight albeit not significant decrease in choline +TMAO…”
Line 343-344: The authors must not make a statement about the effects of NAC and LBP8198 on richness and diversity of intestinal flora. Rather, they have to be more precise and sate that NAC and LBP8191 in combination with choline does not affect intestinal flora compared to choline alone.
Figure 7:
First of all, the readability of Figure 7A-D is poor. The labeling is very small.
Next, the description of the results seems a bit simplified to me and also incomplete (what about Figure 7D, choline LP8189, this is not mentioned at all).
Lines 381-382: Please check the punctuation.
Line 389: “The ratio of Firmicutes/Bacteroides is called F/B ratio, which can be used to evaluate the severity of intestinal flora imbalance.” Please indicate a reference
Line 392: “However, the relative abundance of Firmicutes was slightly decreased in the cho-392 line+NAC, choline+LP8198 group than in the choline group (Fig. 8A)”. This is not true! Please double check, as from the graph it rather looks slightly increased.
Line 393 and follows: Be precise here, in the graph you indicate only a significance for the choline only group, while you in the text state that the F/B ratio is increased in the choline groups compared with to the chow controls. In the same line, I can not detect any significance indicators for the slightly decreased effect of choline+NAC, choline+LP8198.
Discussion, line 404-406: This finding is indeed not really novel. Already in 2017 Chen et al. (PMID: 28871042) related TMAO treatment to increasing ROS levels and inflammatory signaling in HUVECS.
And also the second point, that Cholin inkate induces TMAO associated atherosclerosis and mircrobiota composition has been studied before (Li et al. 2021 PMID: 33863898).
Next, they state in line 410 that, NAC and probiotics relieve abnormal community composition although they state the opposite in line 369.
Line 414-416: The author should also in their animal model in the aortic root measure expression of AMPK, SIRT1 and maybe ROS on order to strengthen their hypothesis.
Line 426: This is maybe a bit overstated. As mentioned above, also Chen et al. 2017 (PMID: 28871042) have linked TMAO to ROS formation. The authors even refer to this study in the following (line 437).
Minor:
Line 40-41: missing space after hepatic (TMA can be oxidized to TMAO 40 by hepatic flavin monooxygenases FMO3 )
Line 125-126: was continued instead of were continuing to fed
Line 188: delete the before inflammatory cytokines
Line 190: inflammatory activation rather than inflammation activation
Line 199: please change inhibited into decreased
Line 485: Please change: “In vivo, this study are consistent with the previous reports” into “In vivo, this study is consistent with the previous reports”
Line 489-490 : “the increase levels of TMAO, TC, 489 HDL-C and LDL-C associated with choline feeding were significantly decreased by NAC 490 and the probiotic LP8198 (Fig.5).” Be more precise here, for instance, LDL-C is not reduced by NAC co-treatment.
Round 2
Reviewer 1 Report
Thank you.
This manuscript is a resubmission of an earlier submission. The following is a list of the peer review reports and author responses from that submission.